# Adherence to antiretroviral therapy and associated factors among Human immunodeficiency virus positive patients accessing treatment at Nekemte referral hospital, west Ethiopia, 2019

**Muktar Abadiga****\*, Tahir Hasen, Getu Mosisa, Eba Abdisa**

School of Nursing and midwifery, Institute of Health Sciences, Wollega University, Nekemte, Ethiopia

* muktarabadiga@gmail.com

## Abstract

**Data Availability Statement:** All relevant data are within the paper and its Supporting Information files.

### Background

Antiretroviral therapy has a remarkable clinical effect in reducing the progress of Acquired Immune Deficiency Syndrome. The clinical outcome of Anti-Retroviral therapy depends on strict adherence. Poor adherence reduces the effectiveness of antiretroviral therapy and increases viral replication. With changes in service delivery over time and differences in socio-demographic status from region to region, it is essential to measure adherence. Therefore, this study aimed to assess adherence to antiretroviral therapy and its associated factors among HIV/AIDS patients accessing treatment at Nekemte referral hospital, West Ethiopia.

### Methods

Institutional based cross-sectional study was conducted on 311 HIV/AIDS patients from March 01 to March 30, 2019. The study participants were selected by a simple random sampling method and interviewed using structured questionnaires. Bivariable logistic regression was conducted to find an association between each independent variable and adherence to antiretroviral medication. Multivariable logistic regression was used to find the independent variables which best predict adherence. The statistical significance was measured using odds ratio at a 95% confidence interval with a p-value of less than 0.05.

### Results

Out of a total of 311 patients sampled, 305 were participated in the study, making a response rate of 98.07%. From these 305 study participants,73.1% (95% CI = 68.2, 78.0) were adherent to their medication. Having knowledge about HIV and its treatment (AOR = 8.24, 95% CI: 3.10, 21.92), having strong family/social support (AOR = 6.21, 95% CI: 1.39, 27.62), absence of adverse drug reaction (AOR = 5.33, 95% CI: 1.95, 14.57), absence of comorbidity of other chronic diseases (AOR = 5.72, 95% CI: 1.91, 17.16) and disclosing HIV

**Funding:** The author received no specific funding for this work.

**Competing interests:** The authors have declared that no competing interests exist.

**Abbreviations:** AIDS, Acquired immunodeficiency syndrome; AOR, Adjusted odd ratio; ART, Antiretroviral therapy; ARV, Anti-retroviral; CI, Confidence interval; EFHAPCO, Ethiopian Federal HIV/AIDS Prevention and Control Office; HIV, Human immunodeficiency; PLWH, People living with HIV; SPSS, Statistical Package for Social Science; WHO, World Health Organization.

status to the family (AOR = 5.08, 95% CI: 2.09, 12.34) were significantly associated with an increased likelihood of adherence to antiretroviral medication.

## Conclusion

The level of adherence to antiretroviral therapy was found low compared to WHO recommendation. The clinician should emphasize reducing adverse drug reaction, detecting and treating co-morbidities early, improving knowledge through health education, and encouraging the patients to disclose their HIV status to their families.

## Background

Human immune deficiency virus (HIV) is one of the most destructive epidemics that continue to be a major global public health issue [1]. Globally, 36.7 million people were living with HIV/ AIDS at the end of 2015 with approximately 70% residing in Sub-Saharan Africa [2]. In Ethiopia, the Federal HIV/AIDS Prevention and Control Office (EFHAPCO) indicates that there are over 718,550 peoples living with HIV/AIDS [3].

Ending HIV/AIDS as a public health threat includes achieving the 90-90-90 target set by WHO to ensure successful lifesaving treatment for millions of people [4]. The aim of 90–90–90 targets is to diagnose 90% of all HIV-positive persons, provide antiretroviral therapy for 90% of those diagnosed, and achieve viral suppression for 90% of those treated by 2020 [5]. This is estimated to result in 73% of people with HIV achieving full viral suppression [6].

Antiretroviral therapy is a lifelong activity to treat human immunodeficiency virus infection [1]. Adherence to medication reflects the extent to which a person's behavior in taking medication corresponds with the recommendation from a health care provider [7]. ART decreases the burden of HIV on patients and prevents the occurrence of opportunistic infections [8 & 9]. The availability of ART has improved the survival rates of HIV patients and reduced HIV related comorbidities [10]. However, strict adherence to antiretroviral therapy is important to decrease the multiplication of the virus [11 & 12]. Different literature showed that above 95% adherence to the therapeutic regimen is required for HIV infected patients to reach full viral suppression [13–16].

The level of adherence to antiretroviral therapy vary from country to country and region to region. The level of adherence to ART is 85.5% in China [17], 84.0% in Myanmar [18], 71.0% in Northern Tanzania [19] and 62.2% in Ghana [20]. In Ethiopia, the rate of adherence to antiretroviral therapy is 88.2% [21]. Poor adherence to antiretroviral therapy reduces the effectiveness of ART and increases drug resistance [22 & 23]. Poor adherence to antiretroviral therapy is also linked to a decrease in CD4 count and higher mortality rates [24]. The studies have shown that factors such as socio-demographic and socio-cultural factors, side effects of ARVs, ART regimes, duration on ART, stress, depression, and anxiety were associated with ART adherence [25 & 26].

Adherence is a dynamic process that changes over time and there has been considerable progress of access to ART and HIV counseling, provision of free ART services, expansion of treatment and increased awareness over the past years. The magnitude and determinants of adherence differ across geopolitical zones with their unique characteristics of culture, economic status, religion, educational status, and health-seeking behaviors. Therefore, with the changes in service delivery and variation in socioeconomic status, it is essential to measure adherence from time to time and in different geographical settings. Since the Nekemte referral

hospital is providing services to more than 3 million peoples and over two thousand ART users, determining the level of ART adherence and identifying its determinants is essential for making appropriate interventions. Therefore, this study was aimed to assess the level of adherence to ART and its associated factors among HIV infected patients accessing treatment at Nekemte referral hospital, western Ethiopia.

## Methods

### Study design and setting

Institutional based cross-sectional study was conducted on HIV/AIDS patients accessing treatment at Nekemte referral hospital from March 01 to March 30, 2019. Nekemte referral hospital is one of the largest hospitals found in the western part of Ethiopia at a distance of 325 kilometers from the capital city Addis Ababa. There were 2251 HIV/AIDS patients on treatment follow up at Nekemte referral hospital at the time of the study. There were 21 medical doctors, 57 nurses, 42 midwives and 13 pharmacists working in this hospital. All ART patients on-treatment follow up at Nekemte referral hospital were the source population. All ART patients who had treatment follow up during the study period were the study population. All patients whose ages are 18 years and above were included in this study.

### Sample size determination and sampling techniques

The sample size was calculated using the formula for estimation of a single population proportion ($n = [(Z\alpha/2)^2 \times P (1-P)]/d^2$) with the assumptions of 95% Confidence Level (CL) and marginal error (d) of 0.05. An adherence level of 0.74 (74.0%) was taken from the study conducted in Addis Ababa [27]. Based on this formula, the calculated sample size yields 296 study participants. After adding a non-response rate of 5% which is 15 study participants, a total of 311 ART patients were enrolled in the study. A simple random sampling using the lottery method was used to select the study participants. The sampling was done using a sampling framework or patient files. First, all 2251 patient cards were collected from the ART clinic and coded. Then, 311 patient cards were randomly selected using lottery methods. Finally, data were collected from randomly selected patients as they came for a hospital visit.

### Measurement and data collection procedure

Data was collected using a structured questionnaire and a face-to-face interview was used for this study. In addition to the interview, other patient information was obtained from patient cards. The dependent variable was adherence to ART and the independent variables were socio-demographic, psychosocial and medication-related characteristics. Oslo 3-item social support scale was used to measure family/social support [28]. The 12 items short version of the HIV stigma scale was used to assess social stigma [29]. The adherence level was measured by counting the number of pills that remain in the patients' bottles when he/she comes for follow up. Then, based on WHO guidelines, patients who reported an intake of $\geq$ 95% of the prescribed medication were considered adherent; and those with a reported intake of $<$ 95% were considered non-adherent [30]. Disease/treatment knowledge was measured using tools which has eight questions taken from the previous similar study [31]. The presence of adverse drug reaction was assessed by asking the patient whether he/she experienced any of them in the last months. Socio-demographic factors, psychosocial and medication-related characteristics were measured as explanatory variables. Data were collected by three trained BSc nurses and one supervisor for a duration of one month.

## Data quality control

The questionnaire was translated to the local language and then back to English by expertise to check for consistency. Five percent (5%) of the questionnaire was pre-tested at Gimbi general hospital and some modifications were made based on the results. The one-day training was given for data collectors. The supervisor was also trained on how to monitor the data collection process.

## Data processing and analysis

The data were coded, checked, cleaned and entered into Epi data version 3.1 and exported to Statistical Package for Social Sciences (SPSS) version 20.0 for analysis. Frequencies, percentages, and other descriptive statistics were done as univariate analysis. Bivariable logistic regression was done to find an association between each independent variable and adherence to antiretroviral medication. All independent variables were entered into multivariable logistic regression for further analysis. Finally, multivariable logistic regression with backward elimination was used to find out the independent variables which best predict patient's adherence to antiretroviral medication. All association and statistical significance were measured using an odds ratio at a 95% confidence interval with a p-value of less than 0.05.

## Ethical consideration

This study was reviewed and approved by the Institutional Review Boards of Wollega University. The purpose of the study was explained to the medical director and staff of the hospital. Written informed consent was obtained from the study participants. The patients' information was also kept confidential.

# Results

## Socio-demographic characteristics of participants

Out of the total of 311 study participants sampled, 305 have participated in the study making a response rate of 98.07%. From a total of 305 participants, 140 (45.9%) were male. One hundred six (34.8%) of the study participants lie in the age group between 29–38 years. The minimum age of the study participants was 18 years and the maximum age was 56 years, with a median of 31.00 years. Concerning marital status, 126 (41.3%) of the study participants were single. The majority of the study participants were Oromo 184 (60.3%) in Ethnicity. Concerning educational status, 123 (40.3%) were completed grades 9–12. Regarding monthly income, 81 (26.6%) gets a monthly income of 1501–2000 Ethiopian birr. (Table 1).

## Clinical and behavioral characteristics of participants

The majority of the study participants, 212 (69.5%) had no co-morbidity of other chronic diseases. Two hundred-one (65.9%) had knowledge about HIV and its treatment. Two hundred seven (67.9%) had disclosed their HIV status to their family. One hundred forty (45.9%) had CD4 count between 200–500 cells. Regarding the stage of HIV, 116 (38.0%) were on stage II and 98 (32.1%) were on stage I. One hundred thirty-eight (45.2%) of the study participants had strong social support. More than half of the study participants, 180 (59.0%) don't experienced social stigma. One hundred ninety-four (63.6%) of the study participants had no history of current substance use. Concerning the disease duration, 115 (37.7%) had a duration of 6–10 years. The majority of the study participants, 191 (62.6%) had no adverse drug reaction. (Table 2).

**Table 1. Distribution of study participants by socio-demographic characteristics among HIV/AIDS patients at Nekemte referral hospital, West Ethiopia, 2019 (n = 305).**

| Variables | Category | Frequency | Percentages |
|---|---|---|---|
| sex | Male | 140 | 45.9 |
| | Female | 165 | 54.1 |
| | Total | 305 | 100 |
| Ethnicity | Oromo | 184 | 60.3 |
| | Amhara | 81 | 26.6 |
| | Tigre | 18 | 5.9 |
| | Gurage | 12 | 3.9 |
| | Others | 10 | 3.3 |
| | Total | 305 | 100 |
| Age | 18–28 | 104 | 34.1 |
| | 29–38 | 106 | 34.8 |
| | 39–48 | 46 | 15.1 |
| | >=48 | 49 | 16.1 |
| | Total | 305 | 100 |
| Marital status | Married | 101 | 33.1 |
| | Single | 126 | 41.3 |
| | Divorced | 40 | 13.1 |
| | Widowed | 38 | 12.5 |
| | Total | 305 | 100 |
| Educational status | No formal education | 20 | 6.6 |
| | Primary school (1–8) | 110 | 36.1 |
| | Secondary (9–12) | 123 | 40.3 |
| | Degree and above | 52 | 17.0 |
| | Total | 305 | 100 |
| Occupational status | Government employee | 35 | 11.5 |
| | Private employee | 96 | 31.5 |
| | Farmer | 83 | 27.2 |
| | Merchant | 81 | 26.6 |
| | Others | 10 | 3.3 |
| | Total | 305 | 100 |
| Average monthly income | <500 | 62 | 20.3 |
| | 500–1000 | 56 | 18.4 |
| | 1001–1500 | 73 | 23.9 |
| | 1501–2000 | 81 | 26.6 |
| | >200 | 33 | 10.8 |
| | Total | 305 | 100 |
| Living companion | Yes | 202 | 66.2 |
| | No | 103 | 33.8 |
| | Total | 305 | 100 |

## Level of adherence to medication among the study participants

The adherence rate was calculated by dividing the number of pills the patient actually taken by the number of tablets patients should have taken multiplied by 100. About 3 (1.0%) of the study participants have adherence rate of < 70%, 7 (2.3%) have adherence rate of 70–79.99%, 12 (3.9%) have adherence rate of 80–89.99%, 60 (19.7%) have an adherence rate of 90–94.99%

**Table 2. Distribution of study participants by clinical and behavioral variables among HIV/AIDS patients at Nekemte referral hospital, West Ethiopia, 2019 (n = 305).**

| Variables | Category | Frequency | Percentages |
|---|---|---|---|
| CD4 cell count | <200 | 42 | 13.8 |
|  | 200–500 | 140 | 45.9 |
|  | 501–800 | 111 | 36.4 |
|  | >800 | 12 | 3.9 |
|  | Total | 305 | 100 |
| Stage of HIV | Stage I | 98 | 32.1 |
|  | Stage II | 116 | 38.0 |
|  | Stage III | 48 | 15.7 |
|  | Stage IV | 43 | 14.1 |
|  | Total | 305 | 100 |
| Family/social support | Strong | 138 | 45.2 |
|  | Moderate | 120 | 39.3 |
|  | Poor | 47 | 15.4 |
|  | Total | 305 | 100 |
| Perceived social stigma | Yes | 125 | 41.0 |
|  | No | 180 | 59.0 |
|  | Total | 305 | 100 |
| Current substance use | Yes | 111 | 36.4 |
|  | No | 194 | 63.6 |
|  | Total | 305 | 100 |
| Duration of disease | <1 year | 47 | 15.4 |
|  | 1–5 years | 104 | 34.1 |
|  | 5–10 years | 115 | 37.7 |
|  | >10 years | 39 | 12.8 |
|  | Total | 305 | 100 |
| Adverse drug reaction | Yes | 114 | 37.4 |
|  | No | 191 | 62.6 |
|  | Total | 305 | 100 |
| Waiting time | <30 minutes | 186 | 61.0 |
|  | >/= 30 minutes | 119 | 39.0 |
|  | Total | 305 | 100 |
| Family disclosure status | Yes | 207 | 67.9 |
|  | No | 98 | 32.1 |
|  | Total | 305 | 100 |
| Comorbidity of other chronic diseases | Yes | 93 | 30.5 |
|  | No | 212 | 69.5 |
|  | Total | 305 | 100 |
| Knowledge about HIV and its treatment | Yes | 201 | 65.9 |
|  | No | 104 | 34.1 |
|  | Total | 305 | 100 |

and 223 (73.1%) have an adherence rate of $\geq$ 95%. Then, based on WHO guidelines, patients who reported an intake of $\geq$ 95% of the prescribed medication were considered adherent and those with a reported intake of < 95% were considered as non-adherent [30]. Accordingly, out of the total of 305 study participants, 223 (73.1%) were adherent to their medication (95%

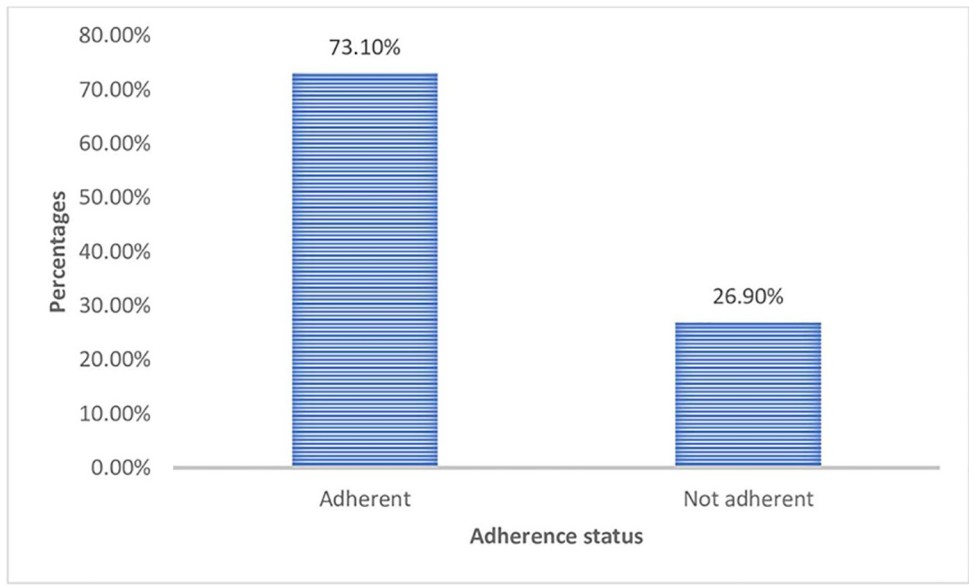

**Fig 1. Bar graph showing the level of adherence to antiretroviral medication among HIV/AIDS patients at Nekemte referral hospital, West Ethiopia, 2019.**

CI = 68.2, 78.0) and 82 (26.9%) were not adherent to their medication (95% CI = 22.0, 31.8). (Fig 1).

## Bivariable logistic regression analysis

In bivariable analysis, socio-demographic characteristics such as age, marital status, income, and a living companion were associated with adherence to the antiretroviral medication. On the other hand, clinical variables such as the side effects of ARV medication, knowledge about HIV and its treatment and comorbidity of other chronic illness were associated with adherence. Behavioral related variables such as family/social support, history of substance use, disclosure of HIV status to family and perceived social stigma were also associated with adherence to the antiretroviral medication at P-value less than 0.05. (Table 3).

## Multivariable logistic regression analysis

In the final model of logistic regression, variables such as adverse drug reaction, social/family support, comorbidity of other chronic diseases, knowledge about HIV and its treatment and disclosure of HIV status to the family were associated with adherence to the antiretroviral medication at P value less than 0.05. Study participants who had strong family/social support were more likely to adhere to their medication than those who had poor family/social support (AOR = 6.21, 95% CI: 1.39, 27.62). Study participants who had knowledge about HIV and its treatment were more likely to adhere to their medication than those who had no knowledge about HIV and its treatments (AOR = 8.24, 95% CI: 3.10, 21.92). Respondents who didn't develop adverse drug reactions were more likely to adhere to their medication than those who developed adverse drug reactions (AOR = 5.33, 95% CI: 1.95, 14.57). Study participants who had no other comorbid chronic diseases were more likely to adhere to their medication than those who had other comorbid chronic illness (AOR = 5.72, 95% CI: 1.91, 17.16). Study

**Table 3. Bivariable logistic regression analysis of factors associated with adherence among HIV/AIDS patients at Nekemte referral hospital, West Ethiopia, 2019 (n = 305).**

| Variables | Adherence | | COR (95%) CI | P value |
|---|---|---|---|---|
| | Adherent (%) | Not adherent (%) | | |
| **Sex** | | | | |
| Male | 97 (69.3%) | 43 (30.7%) | 1 | |
| Female | 126 (76.4%) | 39 (23.6%) | 1.43 (0.86, 2.38) | 0.16 |
| **Age (years)** | | | | |
| 18–28 | 69 (66.3%) | 35 (33.7%) | 2.36 (1.04, 5.36) | 0.04* |
| 29–38 | 78 (73.6%) | 28 (26.4%) | 1.57 (0.73, 3.40) | 0.24 |
| 39–48 | 36 (78.3%) | 10 (21.7%) | 1.08 (0.42, 2.73) | 0.86 |
| >48 | 40 (81.6%) | 9 (18.4%) | 1 | |
| **Marital status** | | | | |
| Married | 81 (80.2%) | 20 (19.8%) | 1 | |
| Single | 92 (73.0%) | 34 (27.0%) | 0.66 (0.35, 1.25) | 0.20 |
| Divorced | 26 (65.0%) | 14 (35.0%) | 0.45 (0.20, 1.03) | 0.06 |
| Widowed | 24 (63.2%) | 14 (36.8%) | 0.42 (0.18, 0.96) | 0.04* |
| **Ethnicity** | | | | |
| Oromo | 134 (72.8%) | 50 (27.2%) | 1 | |
| Amhara | 61 (75.3%) | 20 (24.7%) | 1.13 (0.62, 2.07) | 0.67 |
| Tigre | 13 (72.2%) | 5 (27.8%) | 0.97 (0.32, 2.86) | 0.95 |
| Gurage | 9 (75.0%) | 3 (25.0%) | 1.11(0.29, 4.30) | 0.87 |
| Others | 6 (60.0%) | 4 (40.0%) | 0.56 (0.15, 2.06) | 0.38 |
| **Educational status** | | | | |
| No formal education | 11 (55.0%) | 9 (45.0%) | 0.40 (0.13, 1.20) | 0.10 |
| Primary school (1–8) | 73 (66.4%) | 37 (33.6%) | 0.65 (0.31, 1.38) | 0.26 |
| Secondary school (9–12) | 100 (81.3%) | 23 (18.7%) | 1.44 (0.66, 3.14) | 0.34 |
| College and above | 39 (75.0%) | 13 (25.0%) | 1 | |
| **Residence** | | | | |
| Urban | 62 (68.9%) | 28 (31.1%) | 1 | |
| Rural | 161 (74.9%) | 54 (25.1%) | 1.34 (0.78, 2.31) | 0.28 |
| **Occupation** | | | | |
| Government employee | 28 (80.0%) | 7 (20.0%) | 1 | |
| Private employee | 72 (75.0%) | 24 (25.0%) | 0.75 (0.29, 1.93) | 0.55 |
| Farmer | 56 (67.5%) | 27 (32.5%) | 0.51 (0.20, 1.33) | 0.17 |
| Merchant | 58 (71.6%) | 23 (28.4%) | 0.63 (0.24, 1.64) | 0.34 |
| Others | 9 (90.0%) | 1 (10.0%) | 2.25 (0.24, 20.83) | 0.47 |
| **Income** | | | | |
| <500 EB | 35 (56.5%) | 27 (43.5%) | 0.23 (0.07, 0.67) | 0.008* |
| 500–1000 EB | 41 (73.2%) | 15 (26.8%) | 0.48 (0.15, 1.49) | 0.21 |
| 1001–1500 EB | 51 (69.9%) | 22 (30.1%) | 0.41 (0.14, 1.21) | 0.10 |
| 1501–2000 EB | 68 (84.0%) | 13 (16.0%) | 0.93 (0.30, 2.86) | 0.90 |
| >2000 EB | 28 (84.8%) | 5 (15.2%) | 1 | |
| **Number of pills taken in a day** | | | | |
| 2 tablets | 114 (75.5%) | 37 (24.5%) | 1 | |
| 3 tablets | 52 (72.2%) | 20 (27.8%) | 0.84 (0.44, 1.59) | 0.60 |
| 4 tablets | 43 (69.4%) | 19 (30.6%) | 0.73 (0.38, 1.41) | 0.35 |
| >4 tablets | 14 (70.0%) | 6 (30.0%) | 0.75 (0.27, 2.11) | 0.59 |
| **Living companion** | | | | |
| Yes | 178 (88.1%) | 24 (11.9%) | 9.55 (5.36, 17.0) | 0.000* |

(*Continued*)

**Table 3.** (Continued)

| Variables | Adherence | | COR (95%) CI | P value |
|---|---|---|---|---|
| | **Adherent (%)** | **Not adherent (%)** | | |
| No | 45 (43.7%) | 58 (56.3%) | 1 | |
| **CD4 cell** | | | | |
| <200 | 31 (73.8%) | 11 (26.2%) | 1.40 (0.35, 5.62) | 0.62 |
| 200–500 | 100 (71.4%) | 40 (28.6%) | 1.25 (0.35, 4.38) | 0.72 |
| 501–800 | 84 (75.7%) | 27 (24.3%) | 1.55 (0.43, 5.57) | 0.49 |
| >800 | 8 (66.7%) | 4 (33.3%) | 1 | |
| **HIV stage** | | | | |
| Stage I | 73 (74.5%) | 25 (25.5%) | 1 | |
| Stage II | 88 (75.9%) | 28 (24.1%) | 1.07 (0.57, 2.00) | 0.81 |
| Stage III | 31 (64.6%) | 17 (35.4%) | 0.62 (0.29, 1.31) | 0.21 |
| Stage IV | 31 (72.1%) | 12 (27.9%) | 0.88 (0.39, 1.98) | 0.76 |
| **Side effect of ARV drugs** | | | | |
| Yes | 44 (38.6%) | 70 (61.4%) | 1 | |
| No | 179 (93.7%) | 12 (6.3%) | 23.7 (11.83, 47.57) | 0.000* |
| **Family/social support** | | | | |
| Strong | 134 (97.1%) | 4 (2.9%) | 20.79 (6.54, 66.02) | 0.000* |
| Moderate | 60 (50.0%) | 60 (50.0%) | 0.62 (0.31, 1.23) | 0.17 |
| Poor | 29 (61.7%) | 18(38.3%) | 1 | |
| **Perceived stigma** | | | | |
| Yes | 59 (47.2%) | 66 (52.8%) | 1 | |
| No | 164 (91.1%) | 16 (8.9%) | 11.46 (6.15, 21.35) | 0.000* |
| **Substance use** | | | | |
| Yes | 46 (41.4%) | 65 (58.6%) | 1 | |
| No | 177 (91.2%) | 17 (8.8%) | 14.71 (7.87, 27.47) | 0.000* |
| **Disease duration** | | | | |
| <1 year | 35 (74.5%) | 12 (25.5%) | 0.87 (0.32, 2.36) | 0.79 |
| 1–5 years | 76 (73.1%) | 28 (26.9%) | 0.81 (0.34, 1.98) | 0.64 |
| 6–10 years | 82 (71.3%) | 33 (28.7%) | 0.74 (0.31, 1.74) | 0.49 |
| >10 years | 30 (76.9%) | 9 (23.1%) | 1 | |
| **Treatment duration** | | | | |
| <1 year | 38 (79.2%) | 10 (20.8%) | 1 | |
| 1–5 years | 85 (73.3%) | 31 (26.7%) | 0.72 (0.32, 1.62) | 0.42 |
| 6–10 years | 78 (69.6%) | 34 (30.4%) | 0.60 (0.27, 1.35) | 0.21 |
| >10 years | 22 (75.9%) | 7 (24.1%) | 0.82 (0.27, 2.48) | 0.73 |
| **Knowledge about HIV and its treatment** | | | | |
| Knowledgeable | 181 (90.0%) | 20 (10.0%) | 13.36 (7.29, 24.47) | 0.000* |
| Not knowledgeable | 42 (40.4%) | 62 (59.6%) | 1 | |
| **Waiting time** | | | | |
| <30 minutes | 140 (75.3%) | 46 (24.7%) | 1.32 (0.79, 2.20) | 0.28 |
| >/= 30 minutes | 83 (69.7%) | 36 (30.3%) | 1 | |
| **comorbidity of other chronic illness** | | | | |
| Yes | 35 (37.6%) | 58 (62.4%) | 1 | |
| No | 188 (88.7%) | 24 (11.3%) | 12.98 (7.14, 23.58) | 0.000* |
| **Family disclosure status** | | | | |
| Yes | 184 (88.9%) | 23 (11.1%) | 12.10 (6.68, 21.89) | 0.000* |
| No | 39 (39.8%) | 59 (60.2%) | 1 | |

* shows significant at P-value <0.05

COR: Crude Odd Ratio, CI: Confidence Interval

**Table 4. Multivariable logistic regression analysis of factors associated with adherence among HIV/AIDS patients at Nekemte referral hospital, West Ethiopia, 2019 (n = 305).**

| Variables | Adherence | | AOR (95%) CI | P value |
|---|---|---|---|---|
| | Adherent (%) | Not adherent (%) | | |
| **Family/social support** | | | | |
| Strong | 134 (97.1%) | 4 (2.9%) | 6.21 (1.39, 27.62) | 0.016* |
| Moderate | 60 (50.0%) | 60 (50.0%) | 0.79 (0.26, 2.35) | 0.67 |
| Poor | 29 (61.7%) | 18(38.3%) | 1 | |
| **Knowledge about HIV and its treatment** | | | | |
| Knowledgeable | 181 (90.0%) | 20 (10.0%) | 8.24 (3.10, 21.92) | 0.000* |
| Not knowledgeable | 42 (40.4%) | 62 (59.6%) | 1 | |
| **Side effect of ARV drugs** | | | | |
| Yes | 44 (38.6%) | 70 (61.4%) | 1 | |
| No | 179 (93.7%) | 12 (6.3%) | 5.33 (1.95, 14.57) | 0.001* |
| **comorbidity of other chronic illness** | | | | |
| Yes | 35 (37.6%) | 58 (62.4%) | 1 | |
| No | 188 (88.7%) | 24 (11.3%) | 5.72 (1.91, 17.16) | 0.002* |
| **Family disclosure status** | | | | |
| Yes | 184 (88.9%) | 23 (11.1%) | 5.08 (2.09, 12.34) | 0.000* |
| No | 39 (39.8%) | 59 (60.2%) | 1 | |

* shows significant at P-value <0.05

AOR: Adjusted Odd Ratio, CI: Confidence Interval

participants who disclosed their HIV status to their family were more likely to adhere to their medication than those who don't disclose their HIV status to their family (AOR = 5.08, 95% CI: 2.09, 12.34). (Table 4).

## Discussion

This study examined the level of adherence to ART and associated factors among HIV/AIDS patients in the Nekemte referral hospital, West Ethiopia in March 2019. The overall level of adherence to medication among peoples living with HIV/AIDS in this study is 73.1%. This level of adherence is slightly consistent with the study done in Southern Ethiopia (68.0%) [32], Northern Tanzania (71.0%) [19] and South Africa (69.0%) [33]. However, it is lower than a study done in China (85.5%) [17], Eastern Ethiopia (85.0%) [34], North-West Ethiopia (88.2%) [21], Nepal (87.4%) [35], Myanmar (84.0%) [18], Togo (78.4%) [36] and Indonesia (84.16%) [37]. On the other hand, the levels of adherence to ARV medication in this study is higher than the study done in Ghana (62.2%) [20] and Nyamagana-Mwanza (54.9%) [31]. The difference might be due to variation in sample size, study setting, study design, and study participant's variation. The level of adherence in this study is lower than the world health organization recommendation level [30]. Our current finding showed that adherence to ART is low and emphasis should be placed on counseling the patient on the importance of strict adherence to the medication.

In this study, factors such as social/family support, adverse drug reaction, Knowledge about HIV and its treatment, family disclosure status and comorbidity of other chronic diseases were significantly associated with adherence to ARV medication. Respondents who had strong family/social support were found to be more adherent to their medication than those who have poor family/social support. Support from family and friends has immediate and long-term

positive influences on their adherence. This finding is consistent with the study done in Indonesia [37], Ghana [20], Eastern Ethiopia [34] and Sub-Saharan Africa [38]. However, the study done in China [17], Southern Ethiopia [32], South Africa [33], Nepal [35], Myanmar [18], Togo [36] and North West Ethiopia [21] didn't show a significant association between family/social support and adherence to antiretroviral medication. The presence of support groups is a facilitator for adherence to the medication. It also provides a comfortable environment for sharing experiences and encouragement. Social support boosts the patient's self-esteem and makes easy for the patient to adhere to ART. In contrast, if a patient does not have the support, it becomes difficult for the patient because the situation brings them to be hopeless and be the potential to refuse the treatment.

Respondents who didn't develop adverse drug reactions were more adherent to their medication than those who developed an adverse drug reaction. Adherence to antiretroviral therapy was negatively affected by medication side effects. This might be due to the fact that study participants might skip their medication to avoid drug side effects. This finding is consistent with a study done in Ghana [20], Sub-Saharan Africa [38], Nepal [35], South Africa [33] and Cameroon [39]. However, this finding is not supported by the study done in China [17], Eastern Ethiopia [34], Togo [36], Myanmar [18] and North West Ethiopia [21].

The finding of this study also showed that study participants who had knowledge about HIV and its treatment were more adherent to their medication than those who had no knowledge about HIV and its treatment. This is due to the fact that knowledge of HIV and its treatment might increase their awareness and access to HIV therapy. This finding is consistent with the study done in North West Ethiopia [21], but not supported by many other studies [32, 17, 34, 20, 33, 35 and 18].

In this study, study participants who disclosed their HIV status to their families were more adherent to their medication than who didn't disclose their HIV status to their families. This is similar to a study done in China [17], Tanzania [31], Eastern Ethiopia [34], North West Ethiopia [21], South Africa [33], Myanmar [18] and Togo [36]. When there is no self- disclosure about HIV status, the persons may fear to take their treatments and miss the medication. Disclosing HIV status to families enables PLWH to seek information, express feelings and access support groups. Participants who did not disclose their HIV status put more pressure on themselves and might skip their medication. On the other hand, non-disclosure might impede the participants from obtaining social support.

Respondents who had no comorbidity of other chronic illnesses were more adherent to their medication than those who had comorbidity of other chronic illnesses. The possible reason might be that when the patients had co-morbidities, they might have a pill burden. When the number of pill increases, the patient might experience more adverse effects from the medications which potentially lead them to skip their treatment. This finding is supported by the study done in China [17], Cameroon [39], Eastern Ethiopia [34], Southern Ethiopia [32], and North West Ethiopia [21]. However, this finding is not supported by the study done in Ghana [20] and Northern Tanzania [19]. Unlike other studies done in different parts of the world, age, monthly income, marital status, waiting time, the number of pills and substance use didn't show significant association with adherence in this study.

## Limitation of the study

Since the study was conducted at a single hospital, the results cannot be generalized to people living with HIV/AIDS in Ethiopia. Causality cannot be confirmed since the study design is cross-sectional.

## Conclusion

In this study, the level of adherence to antiretroviral therapy was found low compared to the WHO recommendation. Having knowledge about HIV and its treatment, having strong family/social support, absence of adverse drug reactions, no comorbidity of other chronic diseases and disclosing HIV status to the family were significantly associated with an increased likelihood of adherence to antiretroviral medication. The clinician should focus on adverse drug reactions, early detect and treat co-morbidities, improve knowledge through health education, and encourage HIV/AIDS patients to disclose their HIV status to their families to improve adherence to antiretroviral medication.

## Supporting information

**S1 Questionnaire.**
(DOCX)

**S1 Dataset.**
(SAV)

## Acknowledgments

We would like to acknowledge Nekemte referral hospital medical director and staffs for their cooperation. We are also grateful to the study participants who voluntarily agreed to be interviewed and participated in the study.

## Author Contributions

**Conceptualization:** Muktar Abadiga, Tahir Hasen, Getu Mosisa, Eba Abdisa.

**Data curation:** Muktar Abadiga, Tahir Hasen, Getu Mosisa, Eba Abdisa.

**Formal analysis:** Muktar Abadiga, Tahir Hasen, Getu Mosisa, Eba Abdisa.

**Investigation:** Muktar Abadiga, Tahir Hasen, Getu Mosisa, Eba Abdisa.

**Methodology:** Muktar Abadiga, Tahir Hasen, Getu Mosisa, Eba Abdisa.

**Project administration:** Muktar Abadiga, Tahir Hasen, Getu Mosisa.

**Resources:** Muktar Abadiga, Tahir Hasen, Eba Abdisa.

**Software:** Muktar Abadiga, Getu Mosisa.

**Supervision:** Muktar Abadiga, Eba Abdisa.

**Validation:** Muktar Abadiga, Tahir Hasen, Getu Mosisa, Eba Abdisa.

**Visualization:** Muktar Abadiga, Tahir Hasen, Getu Mosisa, Eba Abdisa.

**Writing – original draft:** Muktar Abadiga, Tahir Hasen, Getu Mosisa, Eba Abdisa.

**Writing – review & editing:** Muktar Abadiga, Tahir Hasen, Getu Mosisa, Eba Abdisa.

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
