## [Decision Letter · Decision Letter 0]

11 Dec 2019

PONE-D-19-27407

Adherence to antiretroviral therapy and associated factors among HIV positive adults attending treatment at Nekemte referral hospital, west Ethiopia, 2019.

PLOS ONE

Dear Dr Muktar Abadiga,

Thank you for submitting your manuscript to PLOS ONE. After careful consideration, we feel that it has merit but does not fully meet PLOS ONE’s publication criteria as it currently stands. Therefore, we invite you to submit a revised version of the manuscript that addresses the points raised during the review process.

Specifically, one of the two reviewers raised very important points which merit to be considered prior to consideration for publication.

We would appreciate receiving your revised manuscript by Jan 25 2020 11:59PM. To enhance the reproducibility of your results, we recommend that if applicable you deposit your laboratory protocols in protocols.io, where a protocol can be assigned its own identifier (DOI) such that it can be cited independently in the future. For instructions see: http://journals.plos.org/plosone/s/submission-guidelines#loc-laboratory-protocols

We look forward to receiving your revised manuscript.

Kind regards,

Joseph Fokam, Ph.D

Academic Editor

PLOS ONE

Journal Requirements:

2. Please include additional information regarding the survey or questionnaire used in the study and ensure that you have provided sufficient details that others could replicate the analyses. For instance, if you developed and/or translated a questionnaire as part of this study and it is not under a copyright more restrictive than CC-BY, please include a copy, in both the original language and English, as Supporting Information.

3. We noticed you have some minor occurrence(s) of overlapping text with the following previous publication(s), which needs to be addressed:

https://doi.org/10.1186/s13104-019-4553-0

https://doi.org/10.1186/s12879-018-3176-8

In your revision ensure you cite all your sources (including your own works), and quote or rephrase any duplicated text outside the Methods section. Further consideration is dependent on these concerns being addressed.

4. Please include your tables as part of your main manuscript and remove the individual files. Please note that supplementary tables (should remain/ be uploaded) as separate "supporting information" files.

Reviewers' comments:

Reviewer's Responses to Questions

**Comments to the Author**

1. Is the manuscript technically sound, and do the data support the conclusions?

Reviewer #1: Yes

Reviewer #2: Partly

2. Has the statistical analysis been performed appropriately and rigorously? 

Reviewer #1: Yes

Reviewer #2: Yes

3. Have the authors made all data underlying the findings in their manuscript fully available?

Reviewer #1: No

Reviewer #2: No

4. Is the manuscript presented in an intelligible fashion and written in standard English?

Reviewer #1: Yes

Reviewer #2: No

5. Review Comments to the Author

Reviewer #1: Background

1 Inclusion of a brief description of key elements of ART services offered at the hospital can make reader understand the context better.

Sample size determination and sampling techniques

1. Brief description of the sampling method would help reader to understand and determine if there were on potential sources of bias that can affect results and interpretation. What was the sampling framework – ART records at the facility or patients were randomly selected as they came for their clinic visit during the study month?

2. 311 patients should be estimated sample size for the study

3. It is not clear if 311 patients represent total number of patients at the facility, in the study design and setting section, “All ART patients on treatment follow up at Nekemte referral hospital was the source population and the sampled ART patients who had treatment follow up during the study period was the study population”, It would be easy to follow the sampling methodology if total number of patients on ART is mentioned and how random sampling was applied.

Ethical consideration

“Filled out questionnaires were carefully handled and all access to results was kept strictly within the author” brief description how records were handled and stored can help the leader to understand instead of just “carefully handle and kept strictly by author”

Results

311 sampled - state out of how many patients in total.

How were social support categories derived?

What substances were used?

WHO clinical staging and CD 4 count levels- explain when these were determined for the patients as they are likely to be influenced by ART. It is not clear if this was the patient’s status at the time of the study or at the time of the ART initiation. Or how recent?

General comment: the author should ensure the grammatical errors are corrected throughout the manuscript.

Reviewer #2: Review of PONE-D-19-27407 : “Adherence to antiretroviral therapy and associated factors among HIV positive adultsattending treatment at Nekemte referral hospital, west Ethiopia, 2019” for PLoS one

General comments

The author addresses a very important public health problem. As already stated by the author, ART adherence is a determining factor as far as the fight towards rolling-back the HIV/AIDS pandemic is concern. In this write-up, the author tries to determine the adherence status of patients on HAART and goes further to identify factors influencing adherence. The write-up is complete with all the required sections but still has some faults. The English in the whole write-up needs to be carefully corrected by a native English speaker or an expert. Some vital information is missing in the background and methodology section. In addition more clarifications are needed in some sections.

Title: Please do replace the word “attending” with “accessing”

Abstract

1. Please have your abstract read by a native English speaker and correct the errors

2. Please do precise in what direction each of the cited factors affects adherence to treatment.

3. What does the author mean by WHO standard? Please change “WHO standard” to “WHO recommendation”.

Background

The background encloses a lot of grammatical errors that need to be addressed carefully. Practically every sentence needs an English correction that I can’t go citing one after the other. This section is poorly organized with a lot of repetitions. I think in this section, the reader will want to know what ART is, and the importance of this treatment, what ART adherence is, current global findings on the status of adherence globally, in Africa and more specifically Ethiopia, what are the consequences of poor ART adherence and the impeding impact on the triple 90 ambitions set by the WHO, what are some of the factors registered in studies to influence ART adherence? And then you end with a paragraph on why the present study was necessary. Please do reorganize this section accordingly.

4. In the last paragraph of your background. “Since most of the Ethiopian setting is resource limited, there are no routine assessments for ART adherence among PLHIV in Ethiopia”. I don’t see any cause effect relationship in the two parts of this statement. In addition, I think in all HIV treatment centers routine follow-up ART adherence is done at the level of the center.

5. The fact that studies are few does not justify your work. Studies might be few but present enough evidence in that domain. I think that with changes in HIV related strategies, changes in the sociodemographic and economic status, with the routine interventions to combat HIV related stigma and discrimination, with the recent adoption of the test-and-treat strategy……etc, evaluative studies to mark progress on the status of ART adherence and possible associated factors (which might have changed over time) are necessary.

6. Please also do reformulate with more precision the objective of this study presented at the end of this section. Please end with a full stop.

Methods

This section lacks some very important precisions and the English errors persist.

7. Nekemte referral hospital. Nothing is said on the treatment center in this hospital. What rational have you for choosing this hospital for your study? How many patients does this center follow up monthly? How many workers found in the center and what are their qualifications? Are all these workers trained?

8. Refusal to participate is an ethical issue and not an exclusion criteria. In addition, were newly diagnosed patients included? What about those who arrived critically ill?

9. How were participants included in this study? I guess that data collectors were stationed at the treatment center. How was your simple random sampling implemented?

10. Were the data collectors workers at the HIV unit? Or did they come to the unit just to do data collection? How were potential eligible participants identified? What strategies were used to reduce stigmatization associated to the new team if they were new?

11. The design used in this study is disturbing except clarifications will come. Were blood samples of the participants collected for work ups (case of CD4 for instance) or was there any documentary or register review? If all was done by interview, how did you verify the information?

12. The procedure of implementation and the data collection procedure is scanty. Were any administrative procedures undertaken? How did data collectors approach eligible participants? Were interviews done behind closed doors? etc

13. The data processing and analysis section is very scanty and cannot be reproduced. The result section presents two p-values….0.25 and 0.05 for which this is not mentioned in this section. How were variables included into the multivariable regression analysis?

14. The data control section logically should be presented before analysis. In addition what was the content of the one day training delivered to data collectors and supervisors? Were supervisors and data collectors given the same training?

15. Clearly state in the ethical consideration section. What type of consent was obtained from participants.

16. The data collection tool should be made available for publication to ease reproducibility.

Results

Please do correct the English in this section. In general, the confidence intervals of the odd ratios are very large indicative of small cell number as far as the considered variables are concerned.

17. Looking at your age distribution, it seems skewed and therefore the best measure of central tendency is the median.

18. How did you clinically stage your participants only by interview?

19. How did you measure knowledge on HIV and treatment. Please clearly state how this was done on the methods section. Who had good knowledge and who poor knowledge?

20. The variable on social support is vague and undefined in methods. Social support from who? Family, friends, or personnel?. Considering that true social support can only be evaluated in case of disclosure of status. Reported information should be very precise

21. Another problem is what you called adverse drug reaction in your questionnaire. Please precise and clearly define all these variables in the methods section.

22. The adherence status in this study is a very important proportion in this study. Please do present this proportion with its 95% confidence interval. In addition, it would be more light to present those with adherence evaluated to be between below 70%, between 70-80%, 80-90%, 90-95%....before those evaluated above 95%. This will help the reader understand the relative adherence level of the treated population.

Discussion

The English needs strengthening. This section is short and really does not put to context the findings in this research. The authors spend time comparing their findings on the ART adherence status to those in other studies and fail to discuss the true meaning and implication of their results. Some of the factors are addressed and others no, what criteria did the author use in choosing which factors to discuss and which not to. Discrepancies are not really well justified

Conclusion

Please reformulate the sentences and correct the English errors

Tables

The results on the tables are well presented and can be followed but these statistical analyses are not well described in the data analysis section. Please correct accordingly. Also define abbreviations like COR,AOR, CI below your tables.

Data availability statement is missing. The author should provide the data sets analysed for the above findings.

6. PLOS authors have the option to publish the peer review history of their article (what does this mean?). If published, this will include your full peer review and any attached files.

Reviewer #1: No

Reviewer #2: Yes: ATEM Bethel Ajong

---

## [Author Response · Author response to Decision Letter 0]

24 Dec 2019

Date: 22/12/2019

To PLOS ONE

Dear Academic Reviewers, 

Subject: Submission of revised Manuscript “Adherence to antiretroviral therapy and associated factors among Human immunodeficiency virus-positive patients accessing treatment at Nekemte referral hospital, West Ethiopia, 2019”. Thank you very much for reviewing our manuscript. We also greatly appreciate the reviewers for their constructive comments and suggestions. We have carried out the revisions that the reviewers requested and revised the manuscript accordingly. We hope the revised version is now suitable for publication and look forward to hearing from you in due course. Point-by-point responses are attached hereafter this page. 

Sincerely,

Muktar Abadiga 

Corresponding author 

Wollega University, Ethiopia

Response to reviewers

Reviewer #1

Background 

1. Inclusion of a brief description of key elements of ART services offered at the hospital can make reader understand the context better.

Response: This is a very important comment and we included this information in the revised version of our manuscript.

Sample size determination and sampling techniques

1. Brief description of the sampling method would help reader to understand and determine if there were on potential sources of bias that can affect results and interpretation. What was the sampling framework – ART records at the facility or patients were randomly selected as they came for their clinic visit during the study month?

Response: There were about 2251 HIV/AIDS patients on treatment follow up at Nekemte referral hospital at time of study. The patients were randomly selected as they came for their hospital visit during the study period. A simple random sampling method was used to select 311 study participants to be involved in the study. These all informations are included in this revised version of our manuscript.

2. 311 patients should be estimated sample size for the study

Response: Yes, 311 patients were the estimated sample size for the study. Initially, a total of 311 study participants were sampled to be participated in the study. Out of these 311 study participants, six participants were eligible for the study but not willing/refused to take part in the study. Therefore, they were excluded from participating in the study, and only 305 study participants were participated in the study. The detail explanations are included in this revised version of this manuscript.

3. It is not clear if 311 patients represent total number of patients at the facility, in the study design and setting section, “All ART patients on treatment follow up at Nekemte referral hospital was the source population and the sampled ART patients who had treatment follow up during the study period was the study population”, It would be easy to follow the sampling methodology if total number of patients on ART is mentioned and how random sampling was applied.

Response: The total number of HIV/AIDS patients having treatment follow up at Nekemte referral hospital was 2251 which is a source population. 311 patients were the estimated sample size for the study after sample size calculation. From this 311 participants, 305 study participants were participated in the study (6 participants) refused to participate. These 305 participants were selected by using simple random sampling techniques as they came for their hospital visit during the study period. These informations are also included in this revised version of this manuscript.

Ethical consideration

“Filled out questionnaires were carefully handled and all access to results was kept strictly within the author” brief description how records were handled and stored can help the leader to understand instead of just “carefully handle and kept strictly by author”

Response: We mean that immediately after the patients’ interview, the principal investigator collects the questionnaire from data collectors and stored where no one can have access to the questionnaires. Therefore, the questionnaire was not exposed to third parties and the patients’ information was kept confidential. This is modified in the revised version of this manuscript.

Results

311 sampled - state out of how many patients in total.

Response: There were about 2251 HIV/AIDS patients on treatment follow up at Nekemte referral hospital at time of study. Out of this 2251 patients, 311 were sampled to be participated in the study. We mistakenly didn’t indicate this total number of patients on treatment follow up in the previous version of our manuscript. But, we indicated in this revised version of our manuscript.

How were social support categories derived?

Response: Oslo 3-item social support scale were used to measure social support. These items are:

1. How many people are so close to you that you can count on them if you have great personal problems?

1. None

2. 1-2

3. 3-5

4. >5

2. How much interest and concern do people show in what you do?

1. None

2. Little

3. Uncertain

4. Some

5. A lot

3. How easy is to get practical help from neighbors if you should need it?

1. Very difficult

2. Difficult

3. Possible

4. Easy

5. Very easy

The OSS-3 scores ranged from 3-14. Oslo 3-item social support scale has three categories: “Poor support” if the range is 3–8, “Moderate support” if the range is 9–11 and “Strong support” if the range is 12–14.

What substances were used?

Response: According to our study, substance use includes the use of substances such as alcohol, illegal drugs, chat, Heroin, cocaine, marijuana, cigarettes and other tobacco products. In this study, the patient is considered as using substance if he/she is currently using any one of the above mentioned substances. The detail explanations are included in this revised manuscript.

WHO clinical staging and CD 4 count levels- explain when these were determined for the patients as they are likely to be influenced by ART. It is not clear if this was the patient’s status at the time of the study or at the time of the ART initiation. Or how recent?

Response: WHO clinical staging of HIV/AIDS is performed each month the patient visit the clinic according to the hospital in which this study was conducted. Therefore, the WHO clinical stages of HIV/AIDS mentioned in this study was taken from the patients card which was determined by health professional working at ART clinic. So, it was the WHO HIV/AIDS stage at the time of study. However, CD4 count is performed every 3 months in this hospital and the most recent CD4 count was taken for this study. 

General comment

 The author should ensure the grammatical errors are corrected throughout the manuscript.

Response: We agree that the previous version of our manuscript has some grammatical and editorial problems. We used free online grammar correction to solve this problem in this revised version of our manuscript. Therefore, the grammar of our revised manuscript is now improved.

Thank you very much!

Reviewer #2

General comments

The author addresses a very important public health problem. As already stated by the author, ART adherence is a determining factor as far as the fight towards rolling-back the HIV/AIDS pandemic is concern. In this write-up, the author tries to determine the adherence status of patients on HAART and goes further to identify factors influencing adherence. The write-up is complete with all the required sections but still has some faults. The English in the whole write-up needs to be carefully corrected by a native English speaker or an expert. Some vital information is missing in the background and methodology section. In addition, more clarifications are needed in some sections.

Response: Thank you very much for your constructive and crucial comments. We agree that the previous version of our manuscript has some grammatical and editorial problems. We used free online grammar correction to solve this problem in this revised version of our manuscript. Therefore, the grammar of our revised manuscript is now improved. The information missed in the previous submission of our manuscript is also corrected in this revised manuscript.

Title: Please do replace the word “attending” with “accessing”

Response: We accepted this comment and replaced our former title with “Adherence to antiretroviral therapy and associated factors among Human immunodeficiency virus positive patients accessing treatment at Nekemte referral hospital, west Ethiopia, 2019”.

Abstract

1. Please have your abstract read by a native English speaker and correct the errors

Response: We agree that the previous version of our manuscript has some grammatical and editorial problems. As we mentioned above, we used free online grammar correction to solve this problem in this revised version of our manuscript. Therefore, the grammar of our revised manuscript is now improved

2. Please do precise in what direction each of the cited factors affects adherence to treatment.

Response: We mistakenly didn’t show the direction of association of different independent factors with adherence in the previous version of our manuscript. So, we showed the direction of association of independent variables with dependent variable in this revised version of our manuscript. It is indicated as follows: Having knowledge about HIV and its treatment (Adjusted odd ratio=8.13, 95% Confidence interval: 3.06, 21.61), strong family/social support (Adjusted odd ratio= 7.36, 95% Confidence interval: 2.07, 26.10), absence of adverse drug reactions (Adjusted odd ratio = 5.62, 95% Confidence interval: 2.11, 14.93), no comorbidity of other chronic diseases (Adjusted odd ratio = 5.46, 95% Confidence interval: 1.86, 16.02) and disclosing HIV status to the family (Adjusted odd ratio=5.27, 95% Confidence interval: 2.20, 12.62) were significantly associated with increased likelihood of adherence to antiretroviral medication among HIV/AIDS patients. The detail description of these associations are included in this revised version of our manuscript.

3. What does the author mean by WHO standard? Please change “WHO standard” to “WHO recommendation”.

Response: We mean WHO recommendation. So, we accepted this comment and changed “WHO standard” to “WHO recommendation” in this revised version of our manuscript.

Background

The background encloses a lot of grammatical errors that need to be addressed carefully. Practically every sentence needs an English correction that I can’t go citing one after the other. This section is poorly organized with a lot of repetitions. I think in this section, the reader will want to know what ART is, and the importance of this treatment, what ART adherence is, current global findings on the status of adherence globally, in Africa and more specifically Ethiopia, what are the consequences of poor ART adherence and the impeding impact on the triple 90 ambitions set by the WHO, what are some of the factors registered in studies to influence ART adherence? And then you end with a paragraph on why the present study was necessary. Please do reorganize this section accordingly.

Response: These are also a very important comments and we addressed them in this revised version of our manuscript.

4. In the last paragraph of your background. “Since most of the Ethiopian setting is resource limited, there are no routine assessments for ART adherence among PLHIV in Ethiopia”. I don’t see any cause effect relationship in the two parts of this statement. In addition, I think in all HIV treatment centers routine follow-up ART adherence is done at the level of the center.

Response: We mean that Ethiopia is one of the developing countries which has inadequate resources. Therefore, routine assessment of ART adherence among PLHIV is not conducted at the hospital level due to resource constraints. However; we decided to remove this statement from the paragraph and we removed it in this revised version of our manuscript.

5. The fact that studies are few does not justify your work. Studies might be few but present enough evidence in that domain. I think that with changes in HIV related strategies, changes in the sociodemographic and economic status, with the routine interventions to combat HIV related stigma and discrimination, with the recent adoption of the test-and-treat strategy……etc, evaluative studies to mark progress on the status of ART adherence and possible associated factors (which might have changed over time) are necessary.

Response: As we said in the previous submission of our manuscript, only a few studies have been done on the adherence status of ART and its determinant factors in Ethiopia and no study was done particularly in this study area (Nekemte referral hospital). Many of the studies conducted on this issues in Ethiopia were conducted in the Southern, Northern and Eastern parts of Ethiopia. None of the studies were conducted in Western parts of the country specifically in Nekemte referral hospital. The following reasons were also considered before conducting this study at Nekemte referral hospital. These are:

i) Availability of enough population to enroll in the study. Nekemte referral hospital is the largest hospital in western part of Ethiopia with 2251 HIV/AIDS patients on treatment follow up. This number of HIV/AIDS patients in this hospital is much higher than any other hospitals found in western part of Ethiopia. So, selection of this hospital enhances availability of enough and eligible participants for the study. 

ii) The second reason for selection of this hospital is proximity and accessibility of this hospital for data collectors, supervisors as well as for principal investigator to monitor the overall research project.

iii) The third reason for selection of this hospital was there were no competing demands or no other studies that would interfere with this study. In other words, there was no participation of study participants in other studies with similar inclusion criteria which might interfere with this study.

iv) The fourth reason was absence of similar studies conducted in the study area. Even though the number of HIV/AIDS patients on treatment follow up in this hospital is high, nothing is known about adherence of HIV/AIDS patients to ART. Therefore, the reasons mentioned above were taken into account to conduct this study in this hospital.

6. Please also do reformulate with more precision the objective of this study presented at the end of this section. Please end with a full stop.

Response: We accepted this comment and reformulated as follows: “This study was aimed to assess adherence status and associated factors among HIV infected patient having treatment follow up at Nekemte referral hospital.”

Methods

This section lacks some very important precisions and the English errors persist.

Response: The methods part is now written with precision and the grammar of our revised manuscript is now improved.

7. Nekemte referral hospital. Nothing is said on the treatment center in this hospital. What rational have you for choosing this hospital for your study? How many patients does this center follow up monthly? How many workers found in the center and what are their qualifications? Are all these workers trained?

Response: Nekemte referral hospital is found in Nekemte town and is one of the largest towns found in western part of Ethiopia at about 325 kilometers from a capital city of Addis Ababa. There were about 2251 HIV/AIDS patients on treatment follow up at Nekemte referral hospital at time of study. There were no similar studies conducted in the study area. Even though the number of HIV/AIDS patients on treatment follow up in this hospital is high, nothing is known about adherence of HIV/AIDS patients to ART. Regarding the number of workers in this hospital, there were 21 medical doctors, 57 nurses, 42 midwives and 13 pharmacist working in this hospital during this study. All the above informations were included in this revised version of our manuscript.

8. Refusal to participate is an ethical issue and not an exclusion criterion. In addition, were newly diagnosed patients included? What about those who arrived critically ill?

Response: Refusal to participate in the study was not considered as an exclusion criterion in our study. We considered it as non-response rate, that is why our response rate is only 98.07 %. But, the way we wrote this in our previous submission was not appropriate and we removed it in this revised version of our manuscript. We haven’t come across with newly diagnosed patients, and no patient was arrived critically ill during the data collection.

9. How were participants included in this study? I guess that data collectors were stationed at the treatment center. How was your simple random sampling implemented?

Response: The data collection was undergone at ART clinic of Nekemte referral hospital and the study participants were randomly selected using lottery methods (simple random sampling using lottery methods). The randomly selected patients were interviewed at isolated office to maintain privacy. The detail explanations were included in this revised version of our manuscript.

10. Were the data collectors’ workers at the HIV unit? Or did they come to the unit just to do data collection? How were potential eligible participants identified? What strategies were used to reduce stigmatization associated to the new team if they were new?

Response: Data collectors were recruited from nurses working in Gimbi general hospital which is found at about 145 km from Nekemte referral hospital. They came to this Nekemte referral hospital just to do data collection. The data collectors were health professionals and they know professional ethics. Therefore, stigmatization might not be the critical issue. In addition, the nurses (data collectors) were recruited from another hospital and they don’t know the patient personally to stigmatize them.

11. The design used in this study is disturbing except clarifications will come. Were blood samples of the participants collected for work ups (case of CD4 for instance) or was there any documentary or register review? If all was done by interview, how did you verify the information?

Response: In this study, data was collected using a structured questionnaire and face-to-face interview was used for data collection. In addition to the interview, other patient informations were obtained from patient cards. For example: Information such as CD4 level and WHO staging of HIV/AIDS were obtained from the patients’ cards. WHO clinical staging of HIV/AIDS is performed each month the patient visits the clinic and was taken from the patients card which was determined by health professional working at ART clinic. CD4 count is performed every 3 months in this hospital and the most recent CD4 count was taken for this study from the patients’ cards. These informations are also included in the methodology section of this revised manuscript.

12. The procedure of implementation and the data collection procedure is scanty. Were any administrative procedures undertaken? How did data collectors approach eligible participants? Were interviews done behind closed doors? Etc

Response: The patients who fulfilled the inclusion criteria were randomly selected (simple random sampling using lottery methods) as they came for their hospital visit during the study period. Then, the randomly selected patients were brought to private room and the interview was conducted behind the closed doors after the consent was received. The patients were interviewed one by one (one patient at a time) in the isolated room to maintain privacy. These informations are also included in the methodology section of this revised manuscript.

13. The data processing and analysis section is very scanty and cannot be reproduced. The result section presents two p-values….0.25 and 0.05 for which this is not mentioned in this section. How were variables included into the multivariable regression analysis?

Response: Bivariate analysis using logistic regression was done to find association between each independent variable with adherence to antiretroviral medication. All independent variables were entered into multivariable logistic regression for further analysis. Finally, multivariable logistic regression with backward elimination was used to find out the independent variables which best predict patient’s adherence to antiretroviral medication. Therefore, P values of 0.25 was used to determine significance at bivariate level and 0.05 was used to determine significance at multivariate level. However; all association and statistical significance were measured using odds ratio at 95% confidence interval and p-value of less than 0.05. These all are illustrated in this revised version of our manuscript.

14. The data control section logically should be presented before analysis. In addition, what was the content of the one-day training delivered to data collectors and supervisors? Were supervisors and data collectors given the same training?

Response: These are also a very important comments and we incorporated the correction in the revised version of this manuscript. The content of the training for data collectors and supervisor are different and given separately for them. “One-day training on questionnaire items, content of the questionnaire, how the data to be collected and the patient approach was given for data collectors. The supervisor was also trained on how to monitor data collection, checking the completeness of the questionnaire immediately after collection and how to handle and store the filled questionnaire.”

15. Clearly state in the ethical consideration section. What type of consent was obtained from participants?

Response: As we indicated in the previous version of our submission, the written informed consent was obtained. This was indicated under the heading ethical consideration as follows: “All participants of the study were provided written consent, clearly stating the objectives of the study and their right to refuse, and written informed consent was obtained from the participants.”

16. The data collection tool should be made available for publication to ease reproducibility.

Response: The questionnaire used in this study includes socidemographic characteristics, psychosocial and medication related characteristics or clinical and behavioral measurement questionnaire, measurement of adherence scale, Oslo 3-item social support scale and Disease/treatment knowledge. We used Pill counting method, which is an objective method for determining adherence in this study. After the identification of the hospital in which this study was conducted, we communicated with the health professionals to inform the HIV/AIDS patients on treatment follow up to bring the medication bottle or strips dispensed during the next subsequent visit. So, patients were asked to bring residual medication at next subsequent appointments. This was done before the actual period of data collection, and the medication bottle or strips dispensed during the previous visit were brought by the patient. The data collectors were also assured that patients didn’t buy/sell or borrow/give medications from/to others to get by the pill count. The number of pills taken is calculated by subtracting the count of the number of pills remaining from the total number of pills dispensed. Then, the adherence rate was calculated by dividing the number of pills the patient actually taken by number of tablets patient should have taken x100. The following are adherence worksheet we used.

Adherence worksheet

Patient code______________

1. Number of tablets dispensed at last visit_______

2. Number of tablets returned at this visit (count the tablets the patient has brought) _____

3. Number of days since last visit_________

4. Number of tablets client takes per day_____________

5. Number of tablets client actually used_________

6. Number of tablets patient should have taken___________

Percentage of adherence = Number of tablets client actually taken divided by umber of tablets patient should have taken x 100

Finally, based on WHO Guideline, Patients who reported an intake of ≥ 95% of the prescribed medication were considered adherent; and those with a reported intake of < 95% were classified as non-adherent.

Disease/treatment knowledge was measured using eight questions taken from previous similar studies: ART reduces HIV related morbidity? ART reduces HIV related mortality? HIV is controlled by ART? does patient trusts the doctor? do you knows how to deal with side effects? do you stops taking ART on side effects without health professionals’ consultation, do you know the effectiveness of ART? and not abiding to ART leads to drug resistance? This ART knowledge score was calculated as a continuous variable by summing the participant’s number of correct responses to 8 statements. One point was awarded for each correct response (Yes or No for correct statement), and zero for each wrong response, with a maximum obtainable correct score of 8 for each respondent. The knowledge score was categorized into two levels indicated by poor knowledge/not knowledgeable (0-4) and good knowledge/knowledgeable (5-8).

Oslo 3-item social support scale were used to measure social support. The OSS-3 scores ranged from 3-14. Oslo 3-item social support scale has three categories: “Poor support” if the range is 3–8, “Moderate support” if the range is 9–11 and “Strong support” if the range is 12–14.

All the questionnaires used in this study are available from the correspondence author on request. We also sent you this tools as a separate file in this revised version of our manuscript.

Results

Please do correct the English in this section. In general, the confidence intervals of the odd ratios are very large indicative of small cell number as far as the considered variables are concerned.

Response: We agree that the previous version of our manuscript has some grammatical and editorial problems. We used free online grammar correction to solve this problem in this revised version of our manuscript. Therefore, the grammar of our revised manuscript is now improved. 

17. Looking at your age distribution, it seems skewed and therefore the best measure of central tendency is the median.

Response: We agree that the age distribution is slightly skewed to the right (positively skewed). In this age distribution, the mean is greater than the median (mean = 32.63 and median = 31.00). As a reviewer mentioned, the best measure of central tendency when the datas are skewed is median. In this study, the minimum age of the study participants was 18 years and maximum age was 56 years. Therefore, the median age of the study participant was 31.00 years with a range of 38 years (56 years-18 years). Therefore, we modified this in the revised version of our manuscript.

18. How did you clinically stage your participants only by interview?

Response: The data collectors didn’t clinically stage the patient by interview for the purpose of this study. As we discussed in the comment number 11, the WHO clinical staging of HIV/AIDS is performed each month the patient visits the clinic and was taken from the patients card which was determined by health professional working at ART clinic. Therefore, it was not determined only by interview, and therefore it was taken from the patients’ cards.

19. How did you measure knowledge on HIV and treatment? Please clearly state how this was done on the methods section. Who had good knowledge and who poor knowledge?

Response: Disease/treatment knowledge was measured using eight questions taken from previous similar studies: ART reduces HIV related morbidity? ART reduces HIV related mortality? HIV is controlled by ART? does patient trusts the doctor? do you knows how to deal with side effects? do you stops taking ART on side effects without health professionals’ consultation, do you know the effectiveness of ART? and not abiding to ART leads to drug resistance? This ART knowledge score was calculated as a continuous variable by summing the participant’s number of correct responses to 8 statements. One point was awarded for each correct response (Yes or No for correct statement), and zero for each wrong response, with a maximum obtainable correct score of 8 for each respondent. The knowledge score was categorized into two levels indicated by poor knowledge/not knowledgeable (0-4) and good knowledge/knowledgeable (5-8). This is briefly discussed in this revised version of our manuscript.

20. The variable on social support is vague and undefined in methods. Social support from who? Family, friends, or personnel? Considering that true social support can only be evaluated in case of disclosure of status. Reported information should be very precise

Response: The social support measured in this study is a support which can be from families, friends, relatives or any other personnel. It was measured by Oslo 3-item social support scale as mentioned above. These items are:

How many people are so close to you that you can count on them if you have great personal problems?

1. None

2. 1-2

3. 3-5

4. >5

How much interest and concern do people show in what you do?

1. None

2. Little

3. Uncertain

4. Some

5. A lot

How easy is to get practical help from neighbors if you should need it?

1. Very difficult

2. Difficult

3. Possible

4. Easy

5. Very easy

Then, the patient is considered having “poor support” if the range is 3–8, “moderate support” if the range is 9–11 and “strong support” if the range is 12–14.

21. Another problem is what you called adverse drug reaction in your questionnaire. Please precise and clearly define all these variables in the methods section.

Response: The presence of adverse drug reaction was assessed by asking the patient whether he/she experienced any of them. If the patient developed at least one of the common side effects of ARV drugs, he/she was considered as having drug side effects. The question was asked as follows: “Have you experienced any of the side effects of ARV medication listed below?”

Loss of appetite Rash or hypersensitivity

Lipodystrophy Fatigue 

Nausea and vomiting Trouble sleeping

Diarrhea Numbness in hands and legs

Mood changes Sign and symptoms of anemia

 If the patient experienced at least one of the side effects listed above, tick yes as the response to the question which indicate the patient has experienced adverse drug reactions. These are clearly described in the method section of this revised version of our manuscript.

22. The adherence status in this study is a very important proportion in this study. Please do present this proportion with its 95% confidence interval. In addition, it would be more light to present those with adherence evaluated to be between below 70%, between 70-80%, 80-90%, 0-95%... before those evaluated above 95%. This will help the reader understand the relative adherence level of the treated population.

Response: We didn’t indicate the adherence proportion with its 95% confidence interval in the first submission of our manuscript. However; we included in this revised version of our manuscript. Based on reviewer’s request, the detail proportion of adherence were explained in this revised version of our manuscript as follows: “The adherence rate was calculated by dividing the number of pills the patient actually taken by the number of tablets patients should have taken x100. About 3 (1.0%) of the study participants have adherence rate of < 70%, 7 (2.3%) of the study participants have adherence rate of 70-79.9%, 12 (3.9%) of the study participants have adherence rate of 80-89.99%, 60 (19.7%) of the study participants have adherence rate of 90-94.99% and 223 (73.1%) of the study participants have adherence rate of >/= 95%. Then, based on WHO guidelines, patients who reported intake of ≥ 95% of the prescribed medication were considered adherent; and those with a reported intake of < 95% were classified as non-adherent. Accordingly, out of the total of 305 study participants, 223 (73.1 %) were adherent to their medication (95% CI = 68.2, 78.0) and 82 (26.9 %) were not adherent to their medication (95% CI = 22.0, 31.8).”

Discussion

The English needs strengthening. This section is short and really does not put to context the findings in this research. The authors spend time comparing their findings on the ART adherence status to those in other studies and fail to discuss the true meaning and implication of their results. Some of the factors are addressed and others no, what criteria did the author use in choosing which factors to discuss and which not to. Discrepancies are not really well justified.

Response: We agree that the discussion part of our first submission was short. In this revised version of our manuscript, we discussed the meaning and implication of the finding in sufficient details. Some of the factors which initially didn’t addressed are discussed in this revised version. Possible justifications for discrepancies are also indicated. 

Conclusion

Please reformulate the sentences and correct the English errors

Response: We reformulated our conclusion, and we hope the grammar errors are corrected in this revised version of our manuscript.

Tables

The results on the tables are well presented and can be followed but these statistical analyses are not well described in the data analysis section. Please correct accordingly. Also define abbreviations like COR, AOR, CI below your tables.

Response: The statistical analysis used in this study were not fully described in the previous version of our manuscript. We well described the statistical analysis used in this revised version of our manuscript. Abbreviations such as COR, AOR, CI were also defined.

Data availability statement is missing. The author should provide the data sets analyzed for the above findings.

Response: The data used during this study are available on request from the corresponding author. We sent you the data sets analyzed for the findings as independent file in the submission of this revised manuscript The data availability statement is also included in this revised version of our manuscript.

Thank you very much! I really appreciate your comments

---

## [Decision Letter · Decision Letter 1]

9 Apr 2020

PONE-D-19-27407R1

Adherence to antiretroviral therapy and associated factors among Human immunodeficiency virus positive patients accessing treatment at Nekemte referral hospital, west Ethiopia, 2019.

PLOS ONE

Dear Mr Abadiga,

Thank you for submitting your manuscript to PLOS ONE. After careful consideration, we feel that it has merit but does not fully meet PLOS ONE’s publication criteria as it currently stands. Therefore, we invite you to submit a revised version of the manuscript that addresses the points raised during the review process.

We would appreciate receiving your revised manuscript by May 24 2020 11:59PM. To enhance the reproducibility of your results, we recommend that if applicable you deposit your laboratory protocols in protocols.io, where a protocol can be assigned its own identifier (DOI) such that it can be cited independently in the future. For instructions see: http://journals.plos.org/plosone/s/submission-guidelines#loc-laboratory-protocols

We look forward to receiving your revised manuscript.

Kind regards,

M Barton Laws

Academic Editor

PLOS ONE

Additional Editor Comments (if provided):

You have responded to most of the reviewers' comments. However, I must ask you to address some important issues of presentation.

The English is comprehensible but there are still a number of grammatical errors and odd word choices. It would be in your interest to have it very carefully proofread and copy edited by someone with native English fluency.

the background section is highly repetitive. You make the same statements multiple times, in several different ways. It could easily be 1/4 as long.

At Line 152 you do not need to give the response rate here. You give it in the results, which is appropriate.

You need to provide references for the Oslo Social Support Scale and the HIV stigma scale

Line 210 It is SPSS Windows, not window. The publisher of the software should be provided in parentheses.

Line 261 "The minimum age of the study participants was 18 years and maximum age was 56 years. Therefore, the median age of the study participant was 31.00 years with a range of 38

263 years (56 years-18 years)." Again, this is repetitive. Please review the manuscript carefully for redundancy and express yourselves concisely.

Line 316. P values of 0.25 are not considered significant. Conventionally, a p value of >.05 is considered the threshold for significance; even so you should point out that you are making multiple comparisons so some of these associations may be spurious. Please remove the asterisks from all values higher than .05.

Line 329: "Strong family/social support were 6.21 times more likely to adhere to their medication than those who had poor family/social support (Adjusted odd ratio= 6.21, 95%

Confidence interval: 1.39, 27.62)". This is not the meaning of an odds ratio. Please simply say "People with strong family social support were more likely to adhere to their medication than those with poor family/social support (AOR = 6.21, C.onfidence interval: 1.39, 27.62) Remove all references to likelihood ratios.

Table 4: You should run and present a parsimonious model without the non-significant variables.

If you can respond to these comments I may not need to send this out again for peer review, but I do not want to burden reviewers with making these points. I know you want to make a good presentation of your work. I believe this is informative and can usefully guide clinical care, but it has to be readable and properly presented.

Reviewers' comments:

Reviewer's Responses to Questions

2. Is the manuscript technically sound, and do the data support the conclusions?

Reviewer #2: Partly

3. Has the statistical analysis been performed appropriately and rigorously? 

Reviewer #2: Yes

4. Have the authors made all data underlying the findings in their manuscript fully available?

Reviewer #2: Yes

5. Is the manuscript presented in an intelligible fashion and written in standard English?

Reviewer #2: No

6. Review Comments to the Author

Reviewer #2: RE-REVIEW of “Adherence to antiretroviral therapy and associated factors among Humanimmunodeficiency virus positive patients accessing treatment at Nekemte referral hospital, west Ethiopia, 2019”

I wish to thank the author for the point by point rebuttal to the comments brought up in the last review. I appreciate the commitment of the author in addressing each of the reviewers’ comments. I am satisfied with most of the explanations given for my questions and modifications done.

I am however sorry that the manuscript in its present condition does not meet all publication criteria for PLOS one. The article is not presented in an intelligible fashion nor is it written in standard English. The message can be very good but the written communication handicaps everything. It is disturbing that grammatical errors persist in the manuscript right from the first statement of the introduction. I beg to differ with the author; no strengthening of the English in the write up was done. It is not the place of the reviewer to correct the English in a manuscript but from evaluation of the language strength in this scientific write up, I think it hides the scientific message the author is trying to pass. That said, I suggest the author contacts an expert to edit the English in this manuscript while he addresses these additional comments.

1. Take off the full stop from your title.

2. The authors in their background still neglect a key public health goal (the triple 90 ambition by 2020) which is threatened by non-adherence. According to this ambition, 90 percent of all people receiving ART will have viral suppression by 2020. Some of the paragraphs in your background are just a repetition of the same thing in other words. Your data was collected in 2019, a paragraph should present this and I think bringing it up in the discussion of the results is also key.

3. The author states in the last paragraph of background “It is essential to assess the number of people living with HIV/AIDS drop out of treatment programmes and factors affecting adherence to ART”. The reason they state is because of few studies…. And he moves on to give the goal of the study. My question is why Nekemte? And why now? What is the difference in the population constitution between other places in Ethiopia and Nekemte which you think could affect their behaviours and therefore adherence? In the last review, I gave you a direction of thinking to clearly justify your research in that place.

4. Please present the study design before the setting of your study.

5. The author states that a simple random sampling method was adopted (by lottery) to included participants in the study. How was this done? Was it done using the patient files? Or how was it done given that patients on follow-up do not visit the facility all at once? I ask this question because I want to be sure each of the 2251 patients followed up in the facility by your random sampling had equal chances of being included in your study.

6. The additional information added on the sample size section is problematic. First of all, the author does not state why he excluded six participants. Secondly, given that a non-response rate of 5% was considered in building up that sample size, a shortage of six participants is not a problem given that 5% of that sample is 15. I suggest you just take that addition off.

7. Include the place of 0.25 in the data analysis section.

8. The limits of the study should emanate from structured discussions and should be presented before the conclusions because conclusions take note of the limits of the results.

7. PLOS authors have the option to publish the peer review history of their article (what does this mean?). If published, this will include your full peer review and any attached files.

Reviewer #2: Yes: ATEM BETHEL AJONG

---

## [Author Response · Author response to Decision Letter 1]

14 Apr 2020

Date: 12/4/2020

To PLOS ONE

Dear Academic editor, 

Subject: Submission of revised Manuscript “Adherence to antiretroviral therapy and associated factors among Human immunodeficiency virus-positive patients accessing treatment at Nekemte referral hospital, West Ethiopia, 2019”. Thank you very much for reviewing our manuscript. We also greatly appreciate the editor and reviewers for their constructive comments and suggestions. We have carried out the revisions that the editor and reviewers requested and revised the manuscript accordingly. We hope the revised version is now suitable for publication and look forward to hearing from you in due course. Point-by-point responses are attached hereafter this page. 

Sincerely,

Muktar Abadiga 

Corresponding author 

Wollega University, Ethiopia

Editor comments

1. The English is comprehensible but there are still a number of grammatical errors and odd word choices. It would be in your interest to have it very carefully proofread and copy edited by someone with native English fluency.

Response: Thank you very much for your important comments. The previous version of our manuscript has a lot of grammatical and editorial problems. All authors intensively reviewed the document and corrected grammatical and editorial problems. We also used free online grammar correction to solve this problem in this revised version of our manuscript. We also contacted an expert to edit the English in this manuscript.

2. The background section is highly repetitive. You make the same statements multiple times, in several different ways. It could easily be 1/4 as long.

Response: We Avoided these repetitions and tried to make background section short and precise in this revised manuscript

3. At Line 152 you do not need to give the response rate here. You give it in the results, which is appropriate.

Response: We removed and mentioned it in the results

4. You need to provide references for the Oslo Social Support Scale and the HIV stigma scale

Response: We provided these references in this revised version of our manuscript (Line 152&153).

5. Line 210 It is SPSS Windows, not window. The publisher of the software should be provided in parentheses.

Response: We modified this in this revised version of our manuscript (Line 171).

6. Line 261 "The minimum age of the study participants was 18 years and maximum age was 56 years. Therefore, the median age of the study participant was 31.00 years with a range of 38 years (56 years-18 years)." Again, this is repetitive. Please review the manuscript carefully for redundancy and express yourselves concisely.

Response: We avoided this repetition in this revised version of our manuscript (Line 219&220).

7. Line 316. P values of 0.25 are not considered significant. Conventionally, a p value of >0.05 is considered the threshold for significance; even so you should point out that you are making multiple comparisons so some of these associations may be spurious. Please remove the asterisks from all values higher than 0.05.

Response: We changed 0.25 to 0.05 and removed the asterisks from all values higher than .05 in this revised manuscript.

8. Line 329: "Strong family/social support were 6.21 times more likely to adhere to their medication than those who had poor family/social support (Adjusted odd ratio= 6.21, 95%

Confidence interval: 1.39, 27.62)". This is not the meaning of an odds ratio. Please simply say "People with strong family social support were more likely to adhere to their medication than those with poor family/social support (AOR = 6.21, Confidence interval: 1.39, 27.62) Remove all references to likelihood ratios.

Response: We removed all references to likelihood ratios and modified throughout the document accordingly.

9. Table 4: You should run and present a parsimonious model without the non-significant variables.

Response: We removed the non-significant variables from the final model in this revised version of our manuscript.

Reviewers' comments:

The article is not presented in an intelligible fashion nor is it written in standard English. The message can be very good but the written communication handicaps everything. It is disturbing that grammatical errors persist in the manuscript right from the first statement of the introduction. I beg to differ with the author; no strengthening of the English in the write up was done. It is not the place of the reviewer to correct the English in a manuscript but from evaluation of the language strength in this scientific write up, I think it hides the scientific message the author is trying to pass. That said, I suggest the author contacts an expert to edit the English in this manuscript while he addresses these additional comments.

Response: Thank you very much for your constructive and crucial comments. The previous version of our manuscript has a lot of grammatical and editorial problems. We used free online grammar correction to solve this problem in this revised version of our manuscript. We also contacted an expert to edit the English in this manuscript. Therefore, the grammar of our revised manuscript is now improved.

1. Take off the full stop from your title.

Response: We removed it in this revised version

2. The authors in their background still neglect a key public health goal (the triple 90 ambition by 2020) which is threatened by non-adherence. According to this ambition, 90 percent of all people receiving ART will have viral suppression by 2020. Some of the paragraphs in your background are just a repetition of the same thing in other words. Your data was collected in 2019, a paragraph should present this and I think bringing it up in the discussion of the results is also key.

Response: We included the triple 90 ambition by 2020 in this revised manuscript. We also avoided repetition. Data collection time was presented in methodology under study design and setting. We also brought it up in the discussion of the results.

3. The author states in the last paragraph of background “It is essential to assess the number of people living with HIV/AIDS drop out of treatment programmes and factors affecting adherence to ART”. The reason they state is because of few studies…. And he moves on to give the goal of the study. My question is why Nekemte? And why now? What is the difference in the population constitution between other places in Ethiopia and Nekemte which you think could affect their behaviours and therefore adherence? In the last review, I gave you a direction of thinking to clearly justify your research in that place.

Response: We gave justification of this study by considering changes in treatment strategies over time and differences in sociodemographic status in this revised manuscript (Line 95-104).

4. Please present the study design before the setting of your study.

Response: We presented as requested in this revised version of manuscript.

5. The author states that a simple random sampling method was adopted (by lottery) to included participants in the study. How was this done? Was it done using the patient files? Or how was it done given that patients on follow-up do not visit the facility all at once? I ask this question because I want to be sure each of the 2251 patients followed up in the facility by your random sampling had equal chances of being included in your study.

Response: The sampling was done using sampling framework or patient files. First, all 2251 patient cards were collected and coded. Then, 311 patient cards were randomly selected using lottery methods. Finally, data was collected from the randomly selected patients as they came for hospital visit. According to the Nekemte referral hospital, all patients on treatment follow up have an appointment of every 30 days. Therefore, all 2251 patients visit the hospital within 30 days. Our data collection period was also 30 days duration from March 01 to March 30, 2019 and each of the patients had equal chance of being included in the study. 

6. The additional information added on the sample size section is problematic. First of all, the author does not state why he excluded six participants. Secondly, given that a non-response rate of 5% was considered in building up that sample size, a shortage of six participants is not a problem given that 5% of that sample is 15. I suggest you just take that addition off.

Response: Refusal to participate in the study was not considered as an exclusion criterion in this study. We didn’t exclude these 6 study participants but we fail to obtain information from them due to refusal to replay at the time of interview. The estimated sample size for the study was 311 patients. From this, 6 participants were eligible but refused to take part in the study and therefore considered as non-response. That is why only 305 patients were included in the analysis.

7. Include the place of 0.25 in the data analysis section.

Response: Since the associations may be spurious, we changed 0.25 to 0.05 throughout the document. All association (both bivariate & multivariate) and statistical significance were measured at p-value of less than 0.05. We also removed the asterisks from all values higher than 0.05 in table of this revised manuscript.

8. The limits of the study should emanate from structured discussions and should be presented before the conclusions because conclusions take note of the limits of the results.

Response: We rearranged in this revised manuscript

Thank you very much!

---

## [Editor Report · Decision Letter 2]

16 Apr 2020

PONE-D-19-27407R2

Adherence to antiretroviral therapy and associated factors among Human immunodeficiency virus positive patients accessing treatment at Nekemte referral hospital, west Ethiopia, 2019.

PLOS ONE

Dear Mr Abadiga,

Thank you for submitting your manuscript to PLOS ONE. After careful consideration, we feel that it has merit but does not fully meet PLOS ONE’s publication criteria as it currently stands. Therefore, we invite you to submit a revised version of the manuscript that addresses the points raised during the review process.

We would appreciate receiving your revised manuscript by May 31 2020 11:59PM. To enhance the reproducibility of your results, we recommend that if applicable you deposit your laboratory protocols in protocols.io, where a protocol can be assigned its own identifier (DOI) such that it can be cited independently in the future. For instructions see: http://journals.plos.org/plosone/s/submission-guidelines#loc-laboratory-protocols

We look forward to receiving your revised manuscript.

Kind regards,

M Barton Laws

Academic Editor

PLOS ONE

Additional Editor Comments (if provided):

You have responded to the reviewer's comments and to mine. I must tell you that while the quality of English is much improved, there are occasional infelicities. These do not harm the readability of the paper so I leave it to you whether to do another round of copy editing. While I do not feel this needs to be sent out for further review, there are two issues I ask you to address.

At Line 84: "Literature showed that above 95% adherence to the therapeutic regimen is required for HIV infected patients to reach full viral suppression [13-16]. The level of adherence to

ART is 85.5% in China [17], 84.0% in Myanmar [18], 71.0% in Northern Tanzania [19] 87 and 62.2% in Ghana [20]. In Ethiopia, the rate of adherence to antiretroviral therapy is 88.2% [21]."

I believe you are confusing two different ideas. In the first sentence, you are referring to the percentage of doses and individual patient takes as prescribed. In the following sentences, it appears you are referring to the percentage of patients who meet this standard. You should clarify this difference.

At line 139: "The sample size was calculated using the formula for estimation of a single population proportion with the assumptions of 95 % Confidence Level (CL) and marginal error (d) of 0.05. An adherence level of 0.74 (74.0 %) was taken from the study conducted inAddis Ababa [27]. After adding a non-response rate of 5%, a total of 311 ART patients were enrolled in the study."

You do not state the result of this power calculation.

If you can clarify these issues I believe the paper will be suitable for publication.

---

## [Author Response · Author response to Decision Letter 2]

16 Apr 2020

Date: 16/4/2020

To PLOS ONE

Dear Academic editor, 

Subject: Submission of revised Manuscript “Adherence to antiretroviral therapy and associated factors among Human immunodeficiency virus-positive patients accessing treatment at Nekemte referral hospital, West Ethiopia, 2019”. We greatly appreciate the editor for these very important comments which improve our manuscript. We have carried out the revisions that the editor requested and revised the manuscript accordingly. We hope the revised version is now suitable for publication and look forward to hearing from you in due course. Point-by-point responses are attached hereafter this page. 

Sincerely,

Muktar Abadiga 

Corresponding author 

Wollega University, Ethiopia

Editor comments:

 I must tell you that while the quality of English is much improved, there are occasional infelicities. These do not harm the readability of the paper so I leave it to you whether to do another round of copy editing. 

Response: As per your request, we copy edited our manuscript to improve the quality of English.

 At Line 84: "Literature showed that above 95% adherence to the therapeutic regimen is required for HIV infected patients to reach full viral suppression [13-16]. The level of adherence to ART is 85.5% in China [17], 84.0% in Myanmar [18], 71.0% in Northern Tanzania [19] 87 and 62.2% in Ghana [20]. In Ethiopia, the rate of adherence to antiretroviral therapy is 88.2% [21]." I believe you are confusing two different ideas. In the first sentence, you are referring to the percentage of doses and individual patient takes as prescribed. In the following sentences, it appears you are referring to the percentage of patients who meet this standard. You should clarify this difference.

Response: We agree with the editor that the first sentence and the second two sentences have different ideas. The first sentence discusses about the percentage of ART doses (>95%) that the patient should take to achieve viral suppression. This sentence is part of the former paragraph (the immediate above paragraph), and it was mistakenly included in the later paragraph. The idea of this sentence goes with the above paragraph, especially the last sentence of a paragraph (Both talks about the level of adherence required to decrease viral multiplication or viral suppression). So, we move up this sentence to the above paragraph in this revised manuscript. 

On the other hand, he second and third sentences discuss about the level of adherence to ART in different countries including Ethiopia. The percentage of adherence to ART mentioned in these sentences (2nd and 3rd sentences) are the actual rate of adherence to ART out of 100% recorded in different studies. Therefore, we are not referring these percentages to the percentage of patients who meet the standard. Rather, it is the rate of adherence to ART out of 100% among the participants included in those mentioned countries/studies.

 At line 139: "The sample size was calculated using the formula for estimation of a single population proportion with the assumptions of 95 % Confidence Level (CL) and marginal error (d) of 0.05. An adherence level of 0.74 (74.0 %) was taken from the study conducted in Addis Ababa [27]. After adding a non-response rate of 5%, a total of 311 ART patients were enrolled in the study." You do not state the result of this power calculation.

Response: The sample size was calculated using the formula for estimation of a single population proportion (n=[(Zα/2)2 ×P (1-P)]/d 2) with the assumptions of 95 % Confidence Level (CL) and marginal error (d) of 0.05. An adherence level of 0.74 (74.0 %) was taken from the study conducted in Addis Ababa.

n=((Z^(α/2) )^2*p*(1-P))/d^2 

 n=((1.96)^2*0.74*(1-0.74))/〖0.05〗^2 =296 

Therefore, the calculated sample size was 296 and after adding 5% of sample size which is 15 participants, the total sample size became 311. These all information are included in this revised version of our manuscript.

Thank you very much once again!

---

## [Editor Report · Decision Letter 3]

21 Apr 2020

Adherence to antiretroviral therapy and associated factors among Human immunodeficiency virus positive patients accessing treatment at Nekemte referral hospital, west Ethiopia, 2019.

PONE-D-19-27407R3

Dear Dr. Abadiga,

We are pleased to inform you that your manuscript has been judged scientifically suitable for publication and will be formally accepted for publication once it complies with all outstanding technical requirements.

With kind regards,

M Barton Laws

Academic Editor

PLOS ONE
---

## [Editor Report · Acceptance letter]

28 Apr 2020

PONE-D-19-27407R3 

Adherence to antiretroviral therapy and associated factors among Human immunodeficiency virus positive patients accessing treatment at Nekemte referral hospital, west Ethiopia, 2019. 

Dear Dr. Abadiga:

I am pleased to inform you that your manuscript has been deemed suitable for publication in PLOS ONE. Congratulations! Your manuscript is now with our production department. 

With kind regards,

on behalf of

Dr. M Barton Laws 

Academic Editor

PLOS ONE